# How can human-centered design build a story-based video intervention that addresses vaccine hesitancy and bolsters vaccine confidence in the Philippines? A mixed method protocol for project SALUBONG

Mark Donald C. Reñosa,[1,2] Jonas Wachinger,[1] Kate Bärnighausen,[1,3] Mila F. Aligato,[2] Jhoys Landicho-Guevarra,[2] Vivienne Endoma,[2] Jeniffer Landicho,[2] Thea Andrea Bravo,[2] Maria Paz Demonteverde,[2] Jerric Rhazel Guevarra,[2] Nicanor de Claro III,[2] Marianette Inobaya,[2] Maya Adam,[4] Rachel P. Chase,[5] Shannon A. McMahon ![ORCID] [1,6]

**Correspondence to**
Dr Shannon A. McMahon;
mcmahon@uni-heidelberg.de

## ABSTRACT

**Introduction** Since the onset of a dengue vaccine controversy in late 2017, vaccine confidence has plummeted in the Philippines, leading to measles and polio outbreaks in early 2019. This protocol outlines a human-centered design (HCD) approach to co-create and test an intervention that addresses vaccine hesitancy (VH) via narrative and empathy with and among families and healthcare workers.

**Methods and analysis** 'Salubong' is a Filipino term that means to welcome someone back into one's life, reinforcing notions of family ties and friendships. We apply this sentiment to vaccines. Following the phases of HCD, guided by a theoretical framework, and drawing from locally held understandings of faith and acceptance, we will conduct in-depth interviews (IDIs) and focus group discussions (FGDs) in rural and urban Filipino communities that witnessed dramatic increases in measles cases in recent years. During qualitative engagements with caretakers, providers, and policymakers, we will collect narratives about family and community perceptions of childhood vaccinations, public health systems and opportunities to restore faith. IDIs and FGDs will continuously inform the development of (and delivery mechanisms for) story-based interventions. Once developed, we will test our co-created interventions among 800 caretakers and administer a VH questionnaire prior to and immediately following the intervention encounter. We will use the feedback gained through the survey and Kano-style questionnaires to further refine the intervention. Considering the data collection challenges posed by the ongoing COVID-19 pandemic, we have developed workarounds to conduct data collection primarily online. We will use systematic online debriefings to facilitate comprehensive participation of the full research team.

**Ethics and dissemination** Ethical approval has been granted by the Institutional Review Board of the Research

### Strengths and limitations of this study

► Project SALUBONG (**S**hared **A**ppraising, **L**ife stories and **U**ncovering, **B**ridging and **O**ptimizing, and **N**avigating and **G**aining) directly responds to calls in the literature for more community-based research on how vaccine hesitancy (VH) can be addressed and how trust in the public health sector can be bolstered.

► Guided by families and communities, we will co-create an empathic intervention that places the health of children, the concerns of parents and the needs of healthcare workers at the centre to support trust in vaccines.

► Findings will inform future vaccine confidence efforts and contribute to broader policy discussions regarding VH in the Philippines and globally.

► Due to the COVID-19 pandemic, we have shifted some of our data collection online, and our study will be among the first in this setting to outline opportunities and pitfalls of remote qualitative research.

Institute for Tropical Medicine (number 2019–44) and Ethical Commission of Heidelberg University, Faculty of Medicine (S-833/2019). Study findings will be disseminated in scientific conferences and published in peer-reviewed journals.

## BACKGROUND

Vaccines are a cost-effective and safe way to prevent millions of deaths annually.[1 2] Although vaccines represent a seminal achievement in terms of mitigating disease, confidence in vaccines has decreased in many countries in recent years.[3] This drop

in confidence has contributed to stagnation or decreases in immunisation rates, which in turn has resulted in outbreaks of previously controlled or domestically eliminated diseases such as measles and polio.[4–7]

In 2019, the WHO included vaccine hesitancy (VH)—the 'delay in acceptance or refusal of vaccines despite availability of vaccination services'[8]—in its list of the top 10 global health threats requiring high-level attention and research.[9] Literature that has sought to tease out the causes of VH emphasises the '5Cs': complacency (regarding the severity of vaccine-preventable illness), constraints (psychological, financial or structural barriers), a lack of confidence (in vaccines and the broader health system), calculation (the degree to which individuals search for information about vaccines) and collective responsibility (a willingness to protect others).[10] More recently, scholars have started to consider the role of trust and unequal power dynamics in undermining vaccine uptake, describing how families have lost trust in the health system and feel that they have no voice in the face of state-mandated decisions or directives about vaccines.[11–14]

Effective and efficient solutions to address VH are urgently needed, not only to mitigate the re-emergence of vaccine-preventable diseases (such as polio or measles[6 15–17]) but also because the development and uptake of an effective vaccine is a cornerstone of controlling the ongoing COVID-19 pandemic.[18 19]

To date, there is limited guidance in terms of how to successfully combat VH, and the guidance available predominantly stems from high-income settings.[20 21] At the individual level, changing people's attitudes about vaccines has proven difficult, and successful interventions are limited.[22] At the governance level, policymakers in high-income countries have considered or enacted laws to limit access rights and to punish those who reject vaccines (on non-medical grounds) by denying unvaccinated children admission to elementary schools and public playgrounds, and charging parents substantial fines.[23] At the health system level, interventions involving medical professionals have considered how to broach the topic of vaccines in a non-judgemental but affirmative manner, how to listen to parents' vaccine decision-making,[24] and how to facilitate vaccination directly through reminders, prompts, or by reducing logistical barriers.[22]

More recent studies also highlight opportunities in terms of video-based vaccine promotion[24] and educational messages in the form of graphic pictures and anecdotes (focusing on the consequences of not getting a child vaccinated).[25] Several studies have employed human-centered design (HCD), an approach to co-develop interventions with end users.[26] This methodology has led to the creation of mobile apps, education materials, provider guidelines, and the redesign of a health facility, all in the interest of bolstering vaccine uptake.[24 27–31] Results of these HCD studies suggest that the approach supports stronger patient and community engagement.[24 27 30]

Relatively little attention has been paid to VH and ways to address VH in low- and middle-income countries (LMICs).[14 21 24 25] This is particularly problematic for at least three reasons: (1) a majority of the world's vaccine-preventable deaths occur in LMICs;[32] (2) public health and immunisation structures in LMICs are insufficiently equipped to address VH while rolling out national immunisation campaigns and fielding other child health challenges; and (3) in the event of an outbreak of a vaccine-preventable disease, survival rates and containment possibilities are markedly reduced in LMICs where poor structural conditions and extreme poverty can exacerbate pre-existing vulnerabilities.[33–35]

The Philippines, an archipelago with a population of more than 105 million, is among the LMICs that are currently experiencing an unprecedented erosion of public trust in childhood vaccinations.[36–38] Vaccine confidence fell from 93% of adults 'strongly agreeing' to the importance of vaccination in 2015 to 32% in 2018.[36] This is reflected in measles vaccination rates of children under 5, which fell from 88% in 2014 to 55% in 2018.[39] Results from a small 2019 study conducted in two urban communities in Manila, the capital of the Philippines, found that 36% of responding parents had hesitated to give at least one vaccine and/or refused at least one vaccine for their children.[40] These sharp declines are associated with a dengue vaccine controversy in 2017 and the ensuing misinformation that eroded faith in vaccine safety.[36–38 41] This erosion led to the country losing its 19-year polio-free status and sparked measles outbreaks across several islands in 2019, with 47 871 cases including 632 deaths (as compared with 2789 cases and 25 reported deaths in 2018).[39 41 42] The Department of Health (DOH) of the Philippines has therefore made it a priority to win back the trust and confidence of the public in vaccination.[43] The DOH—in partnership with the WHO and UNICEF—has strengthened routine immunisation via the launch of door-to-door immunisation campaigns to increase vaccine uptake and to reach unvaccinated children.[16 44] Although recent data have indicated signs of possible recovery, with the proportion of people agreeing to the importance of vaccines increasing, gains in terms of perceived vaccine safety and effectiveness are less substantial.[3] More tailored and innovative initiatives are necessary to regain community trust in vaccines.

## SALUBONG intervention

'Salubong', the moniker for this research, is a Filipino term that means to welcome someone into one's home or life. In the Philippines, the largest predominantly Christian country in Asia, the term Salubong describes a Catholic dramatisation of the resurrected Jesus encountering his mother Mary, which is a central part of the Easter week across the Philippines. This encounter, as conceptualized in much of Filipino culture, culminates in Jesus—if briefly—reuniting with his mother.[45] From this, Salubong has developed into a Filipino tradition that celebrates the beauty of reconnecting with important figures from one's

past. The Salubong tradition is commonly observed at international airports throughout the Philippines, where many Filipinos are waiting for their loved ones to return home after having sought employment abroad (the diaspora has fostered an undertone of longing and anticipation in many Filipino homes).[46 47] The homecoming of a loved one is viewed as a special and festive event accompanied by a warm embrace and sumptuous meal.

In using this term as our project moniker, we aim to signal to communities that unwelcoming vaccines into homes or lives is a current dilemma, like many of life's dilemmas, but in unwelcoming, there is a progression to rewelcoming. Salubong is intertwined with notions of acceptance, compassion and understanding, and sends the message that we will not rely on an intervention that uses the blunt tool of scientific reason or which excludes population perceptions (particularly as this has proven ineffective in relation to VH[25]). Instead, we will employ HCD[48] to codevelop an intervention together with wary families and communities to encourage them to reconsider their views on childhood vaccinations.

Our central thesis is that by drawing from local narratives, designing, refining, and ultimately testing a story-based intervention that bridges caretakers (e.g., parents, other family members and legal guardians), policymakers, healthcare workers (HCWs), community leaders, and community health workers (following the terminology in the Philippines, where small administrative communities are termed *barangays*, termed 'barangay health workers' (BHWs) in the context of this study), we will lay the foundation to build a meaningful campaign that revives faith in vaccines. This foundation will contribute to the sustainable prevention of outbreaks of vaccine-preventable diseases.

### Study objective

The purpose of this study is to understand local perspectives of VH in the Philippines, and to develop and pilot a health promotion intervention to address VH. Following the multiple stages of HCD, we will co-design, develop, and iterate a community-based intervention.[26] The developed intervention will then be piloted to assess effectiveness and acceptability.

Sub-objectives include:
1. To describe caretakers, policymakers, and community perceptions of the public health system and vaccines.
2. To gather narratives regarding childhood vaccination and health facility experiences from caretakers, HCWs, BHWs, and community leaders (including real-life dialogue between families who delay and refuse vaccinations).
3. To design, pilot, refine, and assess the immediate impact of a picture or video-based intervention with caretakers and community stakeholders.

## METHODS AND DESIGN
### Theoretical underpinning

We draw on the theory of planned behaviour (TPB)[49] to acknowledge how control beliefs (whether to vaccinate and the consequences of this decision), attitudes about vaccinations, and normative beliefs (i.e., notions of social responsibility, subjective norms influenced by social or cultural aspects) shape the intention to vaccinate and vaccination uptake. Such normative beliefs are of particular relevance in this context, as vaccination uptake has a distinct social responsibility dimension. High vaccination coverage is required to protect those who cannot (yet) get vaccinated, but the possibility of 'freeloading' exists for those who refuse vaccinations due to the risks for the individual, yet can still benefit from others being vaccinated.

A number of studies have employed the TPB to explain VH, supporting its utility and describing potential starting points for interventions.[50–52] A recent meta-analysis suggests that the TPB can explain over 50% of the variance in intention to vaccinate, with attitudes and normative beliefs being stronger predictors than perceived behavioural control.[53] A 2017 systematic review of trialled vaccine confidence interventions found few interventions with limited efficacy that aimed at changing individuals' attitudes or their awareness of social norms, and instead identified interventions aimed at reducing barriers and therefore increasing behavioural control as a more promising pathway.[22]

At the core of the current VH crisis in the Philippines is a recent vaccination controversy, which heavily affected the trust in HCWs, the health system, and other vaccination stakeholders.[38] We therefore are also informed by the social ecological model (SEM) to understand how vaccination attitudes and behaviours are shaped on individual, interpersonal, organisational, community, and public policy levels.[54] Kumar and colleagues identified factors across all socioecological levels that affect influenza vaccine uptake,[55] and a number of authors have argued for including multiple socioecological levels when considering how to address VH.[56 57] We will draw on the SEM to gain a better understanding of how stakeholders on different levels perceive barriers or facilitators for vaccinations, and how systematic changes or awareness can increase uptake.

Based on these theoretical models, we developed a framework that guides this project (see figure 1). With the objective to increase individual caretakers' intention to vaccinate their child, we aimed to empathetically listen to caretakers as they describe their attitudes, norms and perceived behavioural control. However, in developing this intervention, we acknowledge that factors such as organisational barriers, purveyors of information or social pressures influence vaccination perception and uptake across several levels of the SEM.

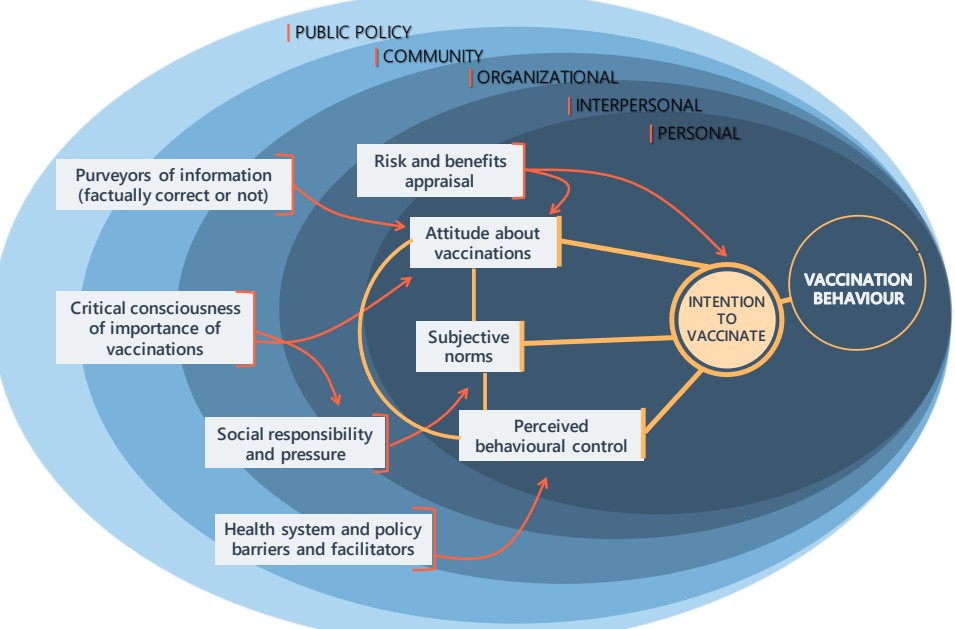

**Figure 1** Preliminary conceptualisation of different influences on vaccination behaviour and vaccine hesitancy to ground intervention development.

### Study setting

The Republic of the Philippines consists of more than 7000 islands divided into 17 administrative regions.[58] This study will be conducted in the Calabarzon region, which has an estimated population of 14 million (with approximately 80% being of Roman Catholic faith and the remaining 20% predominantly belonging to Christian and Muslim denominations), including 1.2 million children under 5 (see figure 2).[59] Calabarzon experienced a 300% increase in measles cases in 2019 as compared to 2018.[39 60 61] The region is middle income and predominantly consists of agriculture, fishing, manufacturing, and high-technology industries.[62 63] Calabarzon is composed of 5 provinces, 20 cities, 123 municipalities, and 4018 *barangays* ('small communities'). The project will be conducted in Dasmariñas City and rural municipalities of Cavite province, which were purposively selected to reflect both rural and urbanised conditions, and to capture varied sociodemographic factors and health facility-related experiences in terms of child health and vaccinations.

### Study population

Our study population will include community leaders (such as the barangay captains and councilors on health), caretakers of young children, policymakers, HCWs (municipal/city health officers, nurses and midwives) and BHWs (community-based health volunteers who help and assist nurses and midwives in the delivery of essential healthcare programmes[64]). Ethnicity, race, political orientation, religion, and class are not criteria for inclusion or exclusion in this study. Caretakers, HCWs, BHWs and community leaders will be eligible to participate in the study if they live within Cavite Province and Dasmariñas City. Participants must be at least 18 years old or an emancipated minor (who are 15–17 years old but with children under 5) to participate. Incapacitated persons are excluded.

### Study design

Informed by HCD,[26] this study employs a mixed-method, exploratory sequential design, drawing on qualitative and then quantitative methods.[65] We will first use qualitative methods (in-depth interviews (IDIs) and focus group discussions (FGDs)) and then quantitative methods (presurvey–postsurveys and Kano questionnaires). Figure 3 shows the summary of the five study phases

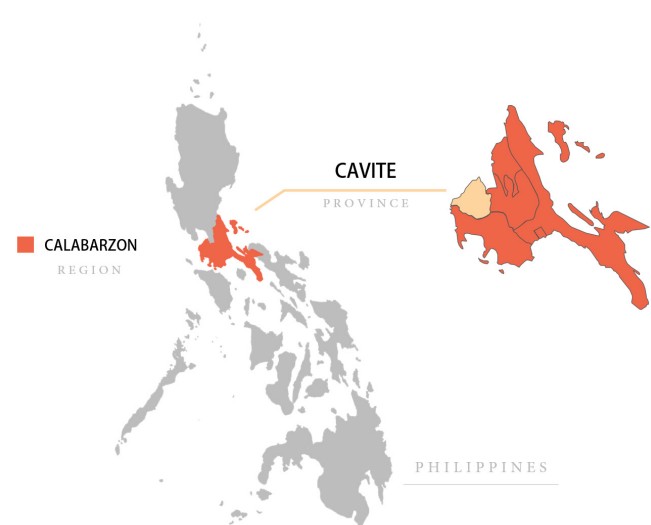

**Figure 2** Location of Cavite, Calabarzon region, included in the project SALUBONG in the Philippines. Map credits: courtesy of https://mapchart.net/.

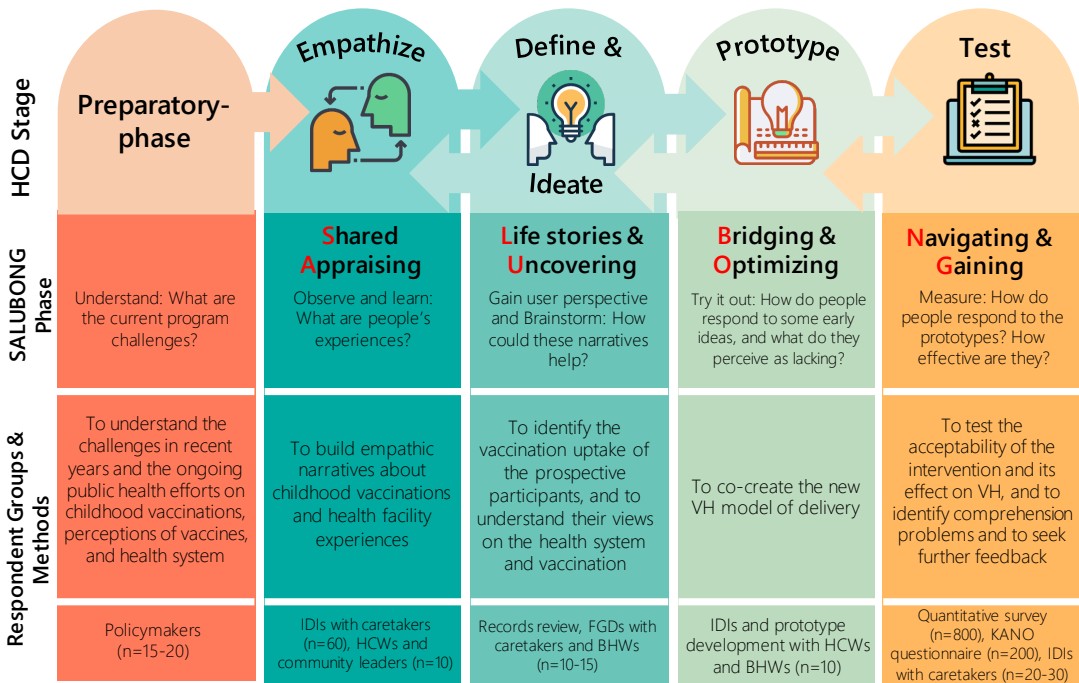

**Figure 3** Summary of the project SALUBONG guided by the HCD. Icon credits: all icons made by Freepik (http://www.freepick.com) courtesy of https://www.flaticon.com/. BHW, barangay health worker; FGD, focus group discussion; HCD, human-centred design; HCW, healthcare worker; IDI, in-depth interview; VH, vaccine hesitancy.

(a preparatory phase followed by the four phases of HCD), along with specific objectives and expected corresponding outputs. Within any given phase, iteration and repetition will typically be necessary. The ultimate goal of the process is to ensure that the fundamental design of a product or programme reflects what users want and works in the setting where they will be using it. Data collection for this study will occur from September 2020 through August 2021.

## Sample sizes and sampling technique

We will partner with HCWs and BHWs who keep lists of caretakers of young children to identify eligible households, particularly families with various experiences or perspectives on vaccines in order to help us capture a range of insights in terms of vaccine attitudes and behaviours.

### Qualitative component

Purposive sampling will be used for the qualitative components of the prephase and phases I–III to gain a maximum range of perspectives and depth of information; sample size estimates are guided by saturation estimates and outlined in figure 3.[66 67] To select the study sites, initially, all municipalities of Cavite Province and districts in Dasmariñas City will be listed. We will select one municipality in Cavite Province and one district in Dasmariñas City with the lowest expanded programme on immunisation coverage for the period of 2018–2019; this approach maximises the probability of finding caretakers who delay or refuse childhood vaccinations. For each selected municipality and district, one to two *barangays*

with the highest number of children under 5 will then be purposively selected.

### Quantitative component

During phase IV, we will rely on household survey data. The estimated sample size per group (intervention and control arms in both rural and urban areas) is 200, which will result in a total of 800 survey responses sought. This will allow us to detect a difference of 15% in the binary outcome between intervention and control groups in each area with an 85% response rate, 5% type I error rate, and 20% type II error rate. To select the study sites, a multistage stratified random sampling frame will be used. The sampling scheme is illustrated in figure 4.

The four *barangays* (two from Dasmariñas City and two from a rural municipality in Cavite) will be purposively selected based on the number of annual births in the most recent report available (preferably 2018–2019) in the rural health unit/city health office. We will select the two barangays with the highest number of births in their respective region and randomly assign study arms.

In each of the selected *barangays*, a list of caretakers with children under 5 will be obtained from the midwife or the BHWs. All caretakers listed will be invited to participate in the study. In the event that the number of interviewed caretakers does not reach the envisioned sample size of 200 per area and study arm, we will continue sampling in the *barangay* with the next most births in the area in the matter described above.

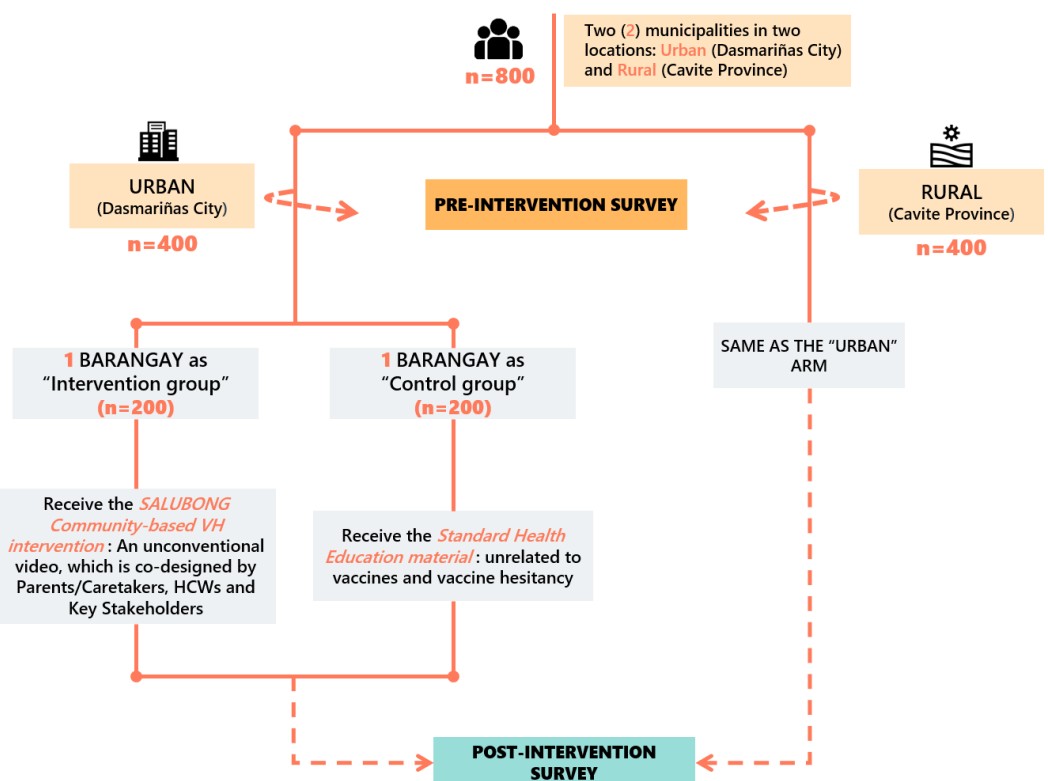

**Figure 4** Study design of phase IV: 'navigating and gaining'. Icon credits: all icons made by Freepik (http://www.freepick.com), courtesy of https://www.flaticon.com/.

## Data collection preparation and data collector training

Before the onset of data collection, we will communicate with DOH officials (national, regional, and provincial offices) to explain study objectives and procedures, and discuss and resolve any concerns regarding data collection. We will also seek their permission and support to conduct the project in the selected primary healthcare facilities. Official communication explaining the nature, study objectives, and procedures will be sent to local government executives and community leaders. Spot maps of the community, if available, will be requested from the rural health units or community health centres. The spot maps will be used to locate prospective families in communities. Courtesy calls to local officials and leaders will be made to seek their support to conduct the project in their respective localities. All required formal endorsements (i.e., memorandums of agreement) will have been granted prior to data collection.

The research team will conduct IDIs, FGDs, and quantitative surveys with the assistance of field interviewers in either English or Filipino, depending on the preference of the participant. Interviewers will be fluent in English and Filipino with bachelor's-level education in nursing, midwifery, or social sciences. Interviewers will be trained for 5 days to collect the data with instruments and online platforms for data collection for this study. Training topics will include modules on VH, interviewing and/or surveying techniques, research ethics, software resources, Kano analysis, and qualitative and quantitative methods.

## Data collection

Data collection will be conducted in five phases: one preparatory phase to develop an understanding of the current situation with regard to vaccination policies and challenges, including but not limited to, the ongoing COVID-19 pandemic, and the four phases of HCD.

### Preparatory phase

We will conduct IDIs with policymakers (n=15–20) to understand the current challenges for ongoing public health efforts on childhood vaccinations, perceptions of vaccines and the health system. We will also explore how the current COVID-19 pandemic poses challenges and opportunities to health education and vaccination efforts.

### Phase I: shared appraising (empathise phase)

This phase of HCD aims at gathering information about how users frame a childhood vaccination problem, how they situate themselves in relation to the problem (probing on sociocultural context), and learning which factors would motivate them to address the problem.

#### IDIs with caretakers of children under 5

We will conduct IDIs (n=60) with caretakers of children under 5. In each selected barangay in Cavite Province and Dasmariñas City, we will purposively select 30 caretakers recommended by a nurse or midwife who will have reviewed vaccination records in order to identify families that have delayed or refused childhood vaccines. Based

on these IDIs, we will build holistic narratives about family perceptions of the public health system and encounters with childhood vaccinations. Participants with particular vaccination or VH experiences will be asked if they are willing to be video-recorded as they tell their personal story related to vaccines. These video-recorded interviews will aid predevelopment of the intervention.

### IDIs with HCWs and community leaders

IDIs (n=10) will also be conducted among HCWs (municipal/city health officers, nurses, midwives) and community leaders (specifically the *barangay* captain and councillors for health) of the selected *barangays* to describe their experiences related to childhood vaccination in their respective community or health facility.

### Refinement of intervention storyboards

Following each interview, the preliminary storyboards developed as a result of the initial IDIs (i.e., 10–15 IDIs) will be presented to participants. Participants (caretakers and HCWs) will be asked to conduct a think-aloud exercise while flipping through these storyboards. After each 10–15 IDIs, or at the end of each week of data collection, the storyboards will be edited and refined based on participants' comments. These revised storyboards will be included in the following round of IDIs for further refinement.

### Phase II: life stories and uncovering (define and ideate phase)

In this phase, end users will suggest ideas to address the problem in collaboration with a research team.

### Records review

From the selected *barangays*, the research team will seek the help of BHWs to identify potential participants based on childhood vaccination records. We will purposively select 50 potential participants in each selected *barangay* of Cavite Province and Dasmariñas City and review their vaccination record through target client lists and individual treatment records. Each case will be allocated to one of three categories:

1. *Fully immunised child*: children who complete one dose of BCG, three doses of oral polio vaccine (OPV), three doses of diptheria–pertussis–tetanus (DPT), three doses of hepatitis B, and one dose of measles vaccine *before* a child's first birthday. These are the children who received their vaccinations within the National Immunisation Programme schedule.
2. *Completely immunised child:* children who receive one dose of BCG, three doses of OPV, three doses of DPT, three doses of hepatitis B, and one dose of measles vaccine *after* a child's first birthday. These are children who have delays in receiving vaccination based on the recommended National Immunisation Programme schedule.
3. *Refusal to vaccinate:* children with the remark 'refused' in the target client lists will be cross-validated with the respective individual treatment records to further review their reasons of refusal.

This process will allow us to initially stratify the participants for FGDs based on their children's vaccination status, preventing contamination and conflict during discussions. Additionally, we will use the records review to confirm delays in children's vaccination schedules, even if these children were later fully immunised.

### Focus group discussions

We will conduct FGDs (n=10–15) with caretakers, stratified (based on the records review and validation of child vaccination cards) into three groups: (1) in favour, (2) delay and (3) refusal of vaccination, to understand how these views evolved or persisted. We will explore the socially held attitudes toward the public health system and vaccines. We will also conduct unstratified FGDs among BHWs of the sampled *barangays* to understand their community and health facility experiences with childhood vaccinations. Results of these FGDs will inform the development of storyboard flip-boards that will be refined in the next phases of the project. To ensure that the design itself of the intervention resonates with the intricate details of Filipino demographic and cultural dynamisms, we will collaborate with local animators with years of groundwork experience in developing interventions in the Philippines who will accompany FGDs and graphically record ideas and concepts in real time, which can be discussed among participants.

### Phase III: bridging and optimising (prototype phase)

In this phase, prototypes and products will be developed and tested in real-world settings with actual users via actual delivery systems.

### IDIs with HCWs and BHWs

The draft intervention, together with data from our FGDs and IDIs, will be presented to HCWs and BHWs. We will conduct 10 IDIs with HCWs and BHWs to get their perspectives and recommendations regarding additional information that needs to be included or refined, namely, related to how, when, and by whom the intervention should be delivered. These IDIs will guide the research team in terms of preferred medium (e.g., paper vs video-based presentation) and favourable delivery approach (e.g., spoken text to accompany the intervention, one-on-one vs group delivery, a stand-alone activity vs nested in existing outreach, etc). The result of this phase will allow us to further refine and finalise the intervention, and to determine the 'point of contact' for the delivery of the intervention in phase IV.

### Phase IV: navigating and gaining (testing phase)

In this phase, the intervention will be introduced to and tested with a larger sample, and large-scale feedback will be sought.

### Pilot testing

We will test the developed intervention versus a control intervention (a standard health education, unrelated to vaccines or VH) in four *barangays* (divided into urban and

rural areas) in the same municipality and district selected in the previous phases. Figure 4 shows the detailed design of the pilot phase. Before and after the delivery of the intervention, the research team will administer short surveys. The preintervention and postintervention surveys (based on best practices for measuring VH[8 68]) will quantify participants' knowledge, attitudes and practices regarding vaccination and VH, and the postintervention survey will additionally assess any changes in knowledge or attitude. We will also collect participants' sociodemographic characteristics (i.e., *barangay* of residence, caretakers' age, sex, civil status, occupation, number of children and education level) and vaccination status of children to allow for further analyses and the identification of potential sampling biases. We will store contact data with an intention to reach out to families in future intervals to determine whether vaccination completion rates differed between intervention and control groups. We will revise the survey tool based on outputs from phases I to III.

### Intervention feedback

After pilot testing, a subset of 200 participants (100 participants randomly sampled from each the urban and rural intervention arms) will be asked a short series of close-ended questions about features of the intervention and feedback on how the intervention made the user feel based on Kano analysis methodology (attractive vs essential).[69] After the Kano survey, 10–15 participants in each area (n=50) will be purposively selected based on critical case sampling and invited to IDIs to identify comprehension problems and to seek further feedback in smaller groups. Since this study only aims to pilot test the developed intervention, the feedback on the intervention will be useful for further refinement before future trials.

### Patient and public involvement

Neither clients nor the public were directly involved in the design, recruitment or conduct of the study; their only involvement is as research participants. However, following the tenets of HCD, the intervention is co-developed with research participants over the entire course of the study. Participants are not involved in initial recruitment, but those participants completing an IDI may be asked to provide contact details of others who might be interested to participate in the study. Participants are not involved in data analysis or the dissemination of findings. The final results will be shared with policymakers, health programmers, and other stakeholders via roundtable discussion, and in the form of articles published in peer-reviewed journals or policy briefs. Research participants will be given access to the final intervention and will be informed through their BHWs and other channels embedded in the community, and have the option to contact study staff for a detailed description of study findings at any time.

### Data collection in times of COVID-19

All IDIs and FGDs were initially planned to be conducted in person. However, the ongoing pandemic and measures taken to curb infection rates pose unique challenges to conducting research. Considering the time-sensitive nature of this project, we have developed new operating procedures for conducting online data collection, to ensure minimal risk for participants and the research team concerning COVID-19, and in compliance with the recommendations by the European Medicines Agency[70] and the Philippines' Inter-Agency Task Force.[71] This includes informed consent processes and the engagement of research team members who are unable to travel to the study site. Until the situation changes, only strictly necessary visits will be performed at sites.

### Remote recruitment and consenting

We will contact prospective participants via email or a phone call to briefly introduce the study aims and invite them to participate. After this initial phone call or email, we will set an appointment for the comprehensive discussion of the study and to answer questions. An information sheet (including study aims, procedures, and expected risks and benefits) and consent forms will be sent to potential participants in advance via email or courier. If participants agree to enrol in the study, we will ask them to sign the consent form during a recorded Skype or Zoom video call, and to take a picture while holding the signed consent form. The signed consent forms will be returned to the research team as scanned or photographed copy, or per courier (as preferred by the participant; courier costs will be covered by the research team). Participants will receive duplicate copies with the signature of the interviewer per courier.

### Focus group discussions

At the time of writing (October 2020), we continue to hope that by the beginning of 2021, when we plan to conduct FGDs, small group discussions in person with appropriate distance will be possible. However, we will conduct remote FGDs if necessary to ensure the safety of participants and research team. In this case, we will draw on the assistance of local stakeholders (HCWs, BHWs, etc) to orient participants on group discussions via a web-based platform.

### Intervention pilot testing

If necessary, amid the pandemic and associated lockdowns, the intervention (VH and control) and surveys (pre and post) will not be delivered in person but instead via an online platform of the participant's choosing. During the presentation of the intervention, there will be no interaction between the researchers and participants to prevent any sort of biases or contamination of the results. We will similarly administer the quantitative survey via an online platform. All online processes will be pilot tested prior to the implementation to assess feasibility and operational challenges. If feasible, the survey

will be in a self-administered format (i.e., interviewers will send the survey link and participants will input their answers in the online form). However, if deemed not feasible, the research team will switch to an interviewer-assisted survey format (ie, interviewers will read the questions to the participants and are responsible in inputting the answers in an online form).

### Systematic debriefings

As a number of research team members will not be able to travel to the study site due to COVID-19 travel restrictions, we will employ remote systematic debriefings[72] to discuss and triangulate findings, to amend interview guides, and to refine lines of inquiry. The weekly debriefings will also allow us to continuously assess, discuss, and refine study tools, data collection procedures, and emerging issues in data collection as a means to ensure fidelity to the tenets of high-quality interviewing. Additionally, in light of the timely relevance of vaccination research in the context of the ongoing pandemic, we will use these debriefings to discuss emerging topics and potential probing approaches with regard to adult and COVID-19 vaccination and continuously refine data collection tools accordingly.

## DATA PROTECTION AND ANALYSIS
### Data quality checks and cleaning process

Most of the data in the project will be primary and collected by the research team. Interviews will be audio-recorded using digital or online audio-recorders. All recorded information will be transcribed verbatim and translated into English. For all qualitative data, the research team collecting and analysing the qualitative data will be directly responsible for quality checks of audio recordings, interview notes, and qualitative transcripts. Random checks of transcripts will be conducted to ensure quality.

All quantitative data will be entered into a customised Microsoft Access data entry system with a built-in validation programme. The data management unit (DMU) of the Research Institute for Tropical Medicine's Department of Epidemiology and Biostatistics will be responsible for quality checks. DMU staff will reach out to the SALUBONG team in the event of questions or data inconsistencies.

### Data storage and protection

All data collected by the research team will be stored according to Germany and Philippines regulations. At the point of data collection, unique codes will be assigned to all participants. A master sheet linking the code with identifying information (including a name and, when feasible, contact information) will be kept in a secure (locked/password-protected) location that is accessible to the investigators. Only the research coordinator will have access to and manage this master sheet. The master sheet will facilitate the process of revoking information

should a participant decide (at a later point) that they wish to have their data removed from the study. Further, audiotaped interviews will also be stored in locked/password-protected computers controlled by the research coordinator. All data will be accessible only to those within the research team but can be made available to others on reasonable request after approval from ethical review boards.

### Data analysis

While the work is divided into phases, the ultimate aim of this study is to analyse and apply findings continuously across phases to address the overall study objective. The principal investigators and coinvestigators will be directly responsible for the triangulation of findings across data sources and the development of an integrated interpretative analysis.

### Qualitative analysis

The qualitative component of this study will be analysed based on constructivist grounded theory as outlined by Charmaz.[73] The application of this approach will help us generate theories on why caretakers would or would not agree to childhood vaccinations, and it will allow us to investigate the constructions and underlying processes of caretakers' perceptions so that informed and relevant approaches can be applied to help resolve highly salient concerns. NVivo Pro V.12 (QSR International Pty) will support qualitative analysis.

### Quantitative analysis

The quantitative component of this study will be analysed using descriptive and inferential approaches. Earlier phases of the study will identify relevant dimensions of knowledge and attitudes to assess in the survey; at its most basic, the analysis will assess binary improvement (1) versus no improvement (0) across those dimensions comparing after and before intervention exposure. Additional analyses will describe and compare the rates of generally favourable versus unfavourable attitudes toward vaccination. Exploratory analyses will seek to uncover sociodemographic variables related to preintervention response patterns as well as postintervention change patterns.

Kano-style questions place user ratings of product or service features on a two-dimensional scale.[69] One dimension is satisfaction, which can range from frustration to delight. The other is functionality, which ranges from not functional to highly effective.[69] Hence, Kano analysis of postintervention survey responses to Kano-style questions will be used to prioritise resource allocation in improving the intervention functionality, with the intention of causally affecting customers' satisfaction. The results will then be used for the revision or redevelopment of intervention features for future, large-scale testing.

### Ethics and dissemination

Ethical approval has been granted by the Institutional Review Board of the Research Institute for Tropical

Medicine (No. 2019–44) and the Ethical Commission of Heidelberg University, Faculty of Medicine (S-833/2019), both recognised ethical review committees. The investigators will consistently respect the principles of ethical research on human subjects described in the Declaration of Helsinki. Written informed consent will be obtained from all eligible participants before data collection begins and after the participants have been fully informed about the study. Study findings will be disseminated in scientific conferences and published in peer-reviewed journals, as well as via policy briefs and round-table discussions. We will adhere to specific reporting guidelines for all publications as applicable, such as the Standards for Reporting Qualitative Research or the Consolidated Criteria for Reporting Qualitative Studies.

## DISCUSSION

Although vaccination uptake and VH are central parts of the current public health and global health discourses, few interventions have proven efficacious in sustainably increasing vaccine confidence. In this study protocol, we explain how we will work with clients, providers and policymakers to develop a narrative-based intervention to increase vaccine confidence in the Philippines. Guided by the tenets of HCD, we will co-develop and iterate this intervention to address the needs and expectations of those it is developed to support.

HCD is a promising new approach in the development of global health interventions.[48] In our application of this methodology, we will move beyond the way it currently is often applied and extend the understanding of 'clients' (following terminology from HCD field understood here as the end users of the intervention) to those involved across several levels including caretakers who are deciding whether to vaccinate their children (clients in the sense of 'target population'), HCWS and BHWs who interact with caretakers (clients in the sense of those who will—or will not—incorporate the intervention into their service delivery), and policymakers and key vaccination stakeholders whose decisions and policies shape the vaccination landscape (clients in the sense of those who decide the implementation of large-scale interventions).

We also expand the existing body of literature on HCD-based vaccine confidence interventions by drawing on a theoretical framework to inform our sampling and design procedures and using well-established behaviour change theories to guide data collection activities. This, for example, includes identifying topics to be probed for in IDIs (e.g., perceived behavioural control about getting a child vaccinated), but also those topics with a clearly social dimension to be focused on in FGDs (e.g., social responsibility of vaccinations). We therefore do not employ a purely inductive approach for conducting qualitative research, as is common in global health qualitative research and HCD approaches, but allow for the existing literature and established theories to guide our approach.

Previous studies demonstrate that decision-making surrounding vaccination is complex, and that those who provide vaccine-related information are in a unique position to sway or break vaccine acceptance.[74 75] A high general trust in the health authorities and in providers who decide what vaccines to introduce is often a deciding factor in vaccine uptake.[76] However, trust and source credibility play a pivotal role not only in direct communication between healthcare providers and their clients, but also across all levels: health-related controversies have been shown to disrupt the trust healthcare providers have in political agents, or implementing government institutions have in those who make the respective policies.[38] Our inclusion of stakeholders from all levels in the design process will support our ability to identify pathways to rebuild both confidence in vaccines and trust in the general system.

The Dengvaxia controversy in the Philippines highlights the pace and reach of misinformation regarding the effectiveness and safety of a vaccine, causing a drop in general vaccine confidence and rolling back decades of immunisation work.[36 38] Dayrit and colleagues[38] stated that the communication gaps among the health authorities themselves, as well as a lack of transparency, proliferation of non-scientific speakers in the mainstream media, and distorted messages in social media played a key role in the panic felt among Filipino families. The Dengvaxia controversy was exacerbated by an exceptionally rapid spread of (mis)information via social media channels.[36 38] In the broader context of vaccination, factually false or misleading information shared in online forums or via social media, often in the form of short quotes or emotionalised images, is considered a central contributor to antivaccination narratives.[77] In many cases, countering these narratives is challenging as it requires longer and complex explanations of vaccination risks and benefits, which are not as easily spread through social media.[78 79] We are taking this into consideration while developing our narrative-based intervention in part or entirely in a form that can be easily relayed (e.g., as a short video or even a GIF). With vaccine confidence or uptake interventions delivered via social media showing promising results,[80 81] we explicitly include this notion in our HCD-based discussion of promising delivery strategies.

The Dengvaxia controversy has been repeatedly linked to the steep decline in vaccination rates and vaccine confidence in the Philippines.[3 36] However, to the best of our knowledge, there so far has not been any qualitative exploration of how this case has shaped narratives about vaccines, health programming, and the health system in general. Furthermore, there has been remarkably little research highlighting Filipino perspectives on how to move forward after the controversy. The SALU-BONG project will thus provide robust, culturally attuned data that can inform future programmes and policies to rebuild trust and confidence in relation to vaccines.[22 82 83]

Finally, similar to many research projects, the ongoing COVID-19 pandemic has forced us to reconsider the way

we collect data to ensure the safety of our participants and research team. We have developed new approaches to remotely undertake this study. In close cooperation with the ethical review boards evaluating this project, we have developed protocols to ensure that recruitment, consent, and rapport building are supported despite the remote nature of data collection. We hope that the procedures outlined in this protocol will spark a discourse on how to conduct ethical and trustworthy research during a pandemic, and that our experiences will allow us to further develop and validate methodological approaches.

**Author affiliations**
[1]Heidelberg Institute of Global Health, Ruprecht Karls Universität Heidelberg, Heidelberg, Germany
[2]Department of Epidemiology and Biostatistics, Research Institute for Tropical Medicine, Muntinlupa City, Philippines
[3]School of Public Health, University of the Witwatersrand, Johannesburg-Braamfontein, Gauteng, South Africa
[4]Department of Pediatrics, Stanford University School of Medicine, Stanford, California, USA
[5]Independent Researcher, Columbus, Ohio, USA
[6]International Health Department, Johns Hopkins University Bloomberg School of Public Health, Baltimore, Maryland, USA

**Acknowledgements** We thank Professor Dr Manuela de Allegri and Dr Veronica Tallo for their expert advice on methodology and local study implementation. Special appreciation is also given to Meghan Obermeyer for her inspiring global health stories, which sparked the idea of this project.

**Contributors** MDCR participated in all aspects of the study design and set-up of local study implementation systems and led the writing of the protocol. JW oversaw the Heidelberg ethics approval process and contributed to and edited the protocol paper. KB participated in the design and writing of the qualitative components and edited the protocol paper. MFA, JL-G, VE, JL, TAB, MPD, JRG and NdC participated in setting up local implementation systems, including development of standard operating procedures for online data collection, and edited the protocol paper. MI advised on the quantitative study design, sample size calculation and local study implementation and edited the protocol paper. MA advised on the HCD process and edited the protocol paper. RC advised on the quantitative study design and sample size calculation and edited protocol paper. SAM supervised all aspects of the study design, the writing and editing of the protocol paper. All authors have read and approved the final version of the manuscript.

**Funding** This study has been externally reviewed and funded following a competitive grant application and review process by the Global Grand Challenges, Bill and Melinda Gates Foundation (OPP1217275). Under the grant conditions of the Foundation, a Creative Commons Attribution 4.0 Generic License has already been assigned to the author accepted manuscript version that might arise from this submission. The funders do not have competing interests and did not play a role in the preparation of this protocol paper.

**Map disclaimer** The depiction of boundaries on the map(s) in this article does not imply the expression of any opinion whatsoever on the part of BMJ (or any member of its group) concerning the legal status of any country, territory, jurisdiction or area or of its authorities. The maps are provided without any warranty of any kind, either express or implied.

**Competing interests** None declared.

**Patient and public involvement** Patients and/or the public were not involved in the design, conduct, reporting or dissemination plans of this research.

**Patient consent for publication** Not required.

**Provenance and peer review** Not commissioned; externally peer reviewed.

**ORCID iD**
Shannon A. McMahon http://orcid.org/0000-0002-7414-7174

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
