## [Reviewer comments · BMJ Open]

ARTICLE DETAILS

TITLE (PROVISIONAL)	How can Human-Centered Design build a story-based video intervention that addresses vaccine hesitancy and bolsters vaccine confidence in the Philippines? A mixed-methods protocol for project SALUBONG
AUTHORS	Reñosa, Mark Donald; Wachinger, Jonas; Bärnighausen, Kate; Aligato, Mila; Landicho-Guevarra, Jhoys; Endoma, Vivienne; Landicho, Jeniffer; Bravo, Thea Andrea; Demonteverde, Maria Paz; Guevarra, Jerric Rhazel; de Claro, Nicanor III; Inobaya, Mariannette; Adam, Maya; Chase, Rachel; McMahon, Shannon

VERSION 1 – REVIEW

REVIEWER	de Figueiredo, Alexandre Universidade Federal da Paraiba, Department of Health Promotion
REVIEW RETURNED	13-Dec-2020

GENERAL COMMENTS	The research topic is relevant for the Philippines and for several other countries that are showing a drop in confidence in national vaccination programs and a consequent increase in the number of cases of diseases that could be prevented by immunization. The protocol is very well prepared and presents a good description of the context and justification for the research. There is a good description of ethical care. The methodology to be used is also well described.
---

REVIEWER	Gowda, Charitha Nationwide Children's Hospital, Infectious Diseases
REVIEW RETURNED	02-Mar-2021

GENERAL COMMENTS	1. Please clarify in the Methods how many barangays in Dasmariñas City and Cavite province (unless a single barangay from each area was selected?) had been selected for the qualitative interventions (IDIs, FDGs). What sampling approach was used to select the barangays? Is it the same approach described in page 10 for the quantitative surveying?2. Will demographic information about the participants (and/or clinical data about the participants' children that may influence vaccination status) be collected in the surveys? This will help ensure that the participants are representative of their communities and/or may help uncover potential biases in the sampling.3. Please clarify if only mothers will be invited to participate or the study will include fathers/other caregivers.4. It appears that authors have categorized VH families as those who have refused at least 1 vaccine. However, this group could be considerably diverse and may warrant consideration of further
--

	substratification (e.g. refusal of specific vaccines such as Measles vaccine, or refusal of all vaccines). 5. Consider including the SRQR checklist with the final completed study.
--	---

REVIEWER	Jamrozik, Euzebiusz University of Oxford, Ethox and the Wellcome Centre for Ethics and Humanities
REVIEW RETURNED	03-Mar-2021

GENERAL COMMENTS	This is an outstanding, timely, and well-targeted research plan. It meets an important need for evidence-based approaches in LMICs that are sensitive to locally-relevant considerations. In particular, meets a critical need to explore how the dengue vaccine controversy has shaped narratives about vaccines & health systems in the Philippines, and this may be relevant to other communities in which such controversies have undermined vaccine confidence (e.g., also related to the dengue vaccine, in Brazil). Other particularly positive aspects of the approach include the identification of communication gaps among key health authorities, the south-south links with previous successes in sub-Saharan Africa, and the plan for public involvement whereby the intervention is co-developed with research participants (i.e., members of the public) Suggestions:  - It might be worth considering allowing for prospective collection of qualitative data regarding adult vaccination (given that this might be important for COVID19, dengue, etc.), as well as child vaccination - even if the latter is the primary focus of the research. Minor issues:  - Use of 'eradication' in the introduction: neither measles nor polio has been eradicated, although on standard definitions both have been eliminated from particular geographic areas.
---

REVIEWER	Lavoie, Kim Montreal Behavioural Medicine Centre, Psychology, UQAM
REVIEW RETURNED	05-Apr-2021

GENERAL COMMENTS	Summary This was an original, timely protocol paper describing the methods for developing a human-centered intervention to address vaccine hesitancy in the Philippines. This paper was very well written, with a compelling rationale and novel, appropriate, systematic methodological approach for co-developing a intervention with relevant stakeholders. This study has a number of important strengths, including being exceptionally timely (in the context of a worldwide pandemic that has brought vaccine hesitancy to the forefront of public health concerns). This study was exceptionally well reasoned: communicable diseases have been shown to disproportionately affect LMIC's and vaccine hesitancy has been an important problem in the Philippines in particular. Public trust has been eroded, resulting in outbreaks polio and measles.
--

	The SALUBONG intervention design is rooted in evidence-based behavioural theories (Theory of Planned Behaviour and the Social-Ecological Model), and builds upon long-held local values and customs focused on acceptance, compassion and understanding. The intervention is community-based and being co-developed in collaboration with all key stakeholders (families, healthcare providers, policymakers), which should increase acceptability, uptake and impact. Objectives are clearly defined: they wish to develop and pilot a health promotion intervention to address vaccine hesitancy in the Philippines that has the potential to be exported to other settings. Methods are clearly described, systematic, and appropriate. The design includes 5 phases: (1) a preparatory phase that involves conducting interviews with policymakers to understand current challenges; (2) phase 1 (empathize) that involves conducting interviews with hesitant parents of children under 5 (using purposive sampling) and with healthcare workers and community leaders to understand their experiences with childhood vaccination; (3) phase 2 (define and ideate) that involves conducting a records review to identify different categories of participants, followed by focus groups; (4) phase 4 (prototype) which involves developing the first iteration of the intervention; and (5) phase 5 (testing), which involves pilot testing the intervention with a larger sample. Methods described at each phase were well defined and appropriate. The authors will use established qualitative methods and analytic strategies, and intervention development stages are in line with established behavioural intervention development models (e.g., MRC, ORBIT). The study populations were well reasoned and carefully selected, and recruitment appears feasible (even in the context of COVID-19). Both qualitative and quantitative analysis methods were well described and appropriate. Sample size estimates were provided and were well justified. Methods are in place to ensure confidentiality and data quality. The integrated knowledge translation approach that involves all relevant stakeholders is evident throughout the project, and should increase acceptance, uptake and efficacy. This study protocol serves as a model for other public health scientists working to change behaviour on a population scale. Suggestions for improvement or clarification:  1. Some abbreviations are not known and were difficult to follow. Suggest using less abbreviations for less common terms (IDI's, etc). 2. The authors will contact Department of Health officials to plan aspects of data collection – it is not known if they have already secured their collaboration. This would be important to confirm. 3. Methods are in place to carefully select qualified interviewers who will undergo training in the protocol. However, methods for assessing the success of training and interview fidelity (once the study is running) to ensure that interviewers are respecting interview protocols. This should be added. 4. The authors are encouraged to check the Equator Network page to verify if this study needs to adhere to any reporting protocols: https://www.equator-network.org/
--	---

VERSION 1 – AUTHOR RESPONSE

Reviewer 1: Dr. Alexandre de Figueiredo Comments	
The research topic is relevant for the Philippines and for several other countries that are showing a drop in confidence in national vaccination programs and a consequent increase in the number of cases of diseases that could be prevented by immunization. The protocol is very well prepared and presents a good description of the context and justification for the research. There is a good description of ethical care. The methodology to be used is also well described.	We thank the reviewer for highlighting that our work is relevant, our protocol is well prepared, our methods are well justified, and our approach is ethical.
Reviewer 2: Dr. Charitha Gowda Comments	
Please clarify in the Methods how many barangays in Dasmariñas City and Cavite province (unless a single barangay from each area was selected?) had been selected for the qualitative interventions (IDIs, FDG)s. What sampling approach was used to select the barangays? Is it the same approach described in page 10 for the quantitative surveying?	We have revisited the original text, and made edits to clarify, as follows: “We will select one municipality in Cavite Province and one district in Dasmariñas City with the lowest Expanded Program on Immunization coverage for the period of 2018-2019; this approach maximizes the probability of finding caretakers who delay or refuse childhood vaccinations. For each selected municipality and district, one to two barangays with the highest number of children under-five will then be purposively selected.”
Will demographic information about the participants (and/or clinical data about the participants' children that may influence vaccination status) be collected in the surveys? This will help ensure that the participants are representative of their communities and/or may help uncover potential biases in the sampling.	Based on your feedback, we have included additional details to bolster clarity, as follows: “We will also collect participants' socio-demographic characteristics (i.e., barangay of residence, caretakers' age, sex, civil status, occupation, number of children and education level) and vaccination status of children to allow for further analyses and the identification of potential sampling biases.”
Please clarify if only mothers will be invited to participate or the study will include fathers/other caregivers.	We clarified and made edits to highlight that we will invite not only mothers but also include other caretakers (e.g., fathers, grandparents). Please see edits throughout the revised manuscript.
It appears that authors have categorized VH families as those who have refused at least 1 vaccine. However, this group could be considerably diverse and may warrant consideration of further substratification (e.g. refusal of specific vaccines such as Measles vaccine, or refusal of all vaccines).	We agree with the reviewer that our categorization of VH families is rather broad. However, in light of the existing data outlining a general decrease in vaccine confidence, the lack of literature on potential substratification in this setting, and the aim of this study to openly co-develop the intervention together with end-users, we decided to make no a-

	priori assumptions regarding the most prominent facets of VH. However, the diversity referred to by the reviewer will be acknowledged and carefully probed on in all data collection activities.
Consider including the SRQR checklist with the final completed study.	We highly appreciate this recommendation; hence, we added details as follows in the 'Ethics and Dissemination' section: “We will adhere to specific reporting guidelines for all publications as applicable, such as the Standards for Reporting Qualitative Research (SRQR) or the Consolidated Criteria for Reporting Qualitative Studies (COREQ).”
Reviewer 3: Euzebiusz Jamrozik Comments	
This is an outstanding, timely, and well-targeted research plan. It meets an important need for evidence-based approaches in LMICs that are sensitive to locally-relevant considerations. In particular, meets a critical need to explore how the dengue vaccine controversy has shaped narratives about vaccines & health systems in the Philippines, and this may be relevant to other communities in which such controversies have undermined vaccine confidence (e.g., also related to the dengue vaccine, in Brazil). Other particularly positive aspects of the approach include the identification of communication gaps among key health authorities, the south-south links with previous successes in sub-Saharan Africa, and the plan for public involvement whereby the intervention is co-developed with research participants (i.e., members of the public).	We are grateful for the reviewer’s observations and thoughtful feedback. We share the reviewer’s scientific interest in capturing narratives and learning how controversies re-shape perspectives.
It might be worth considering allowing for prospective collection of qualitative data regarding adult vaccination (given that this might be important for COVID19, dengue, etc.), as well as child vaccination - even if the latter is the primary focus of the research.	We thank the reviewer for pointing out this particularly timely issue. Considering the recent rapid decline in childhood vaccination rates in the country with regards to established vaccines (as compared to novel vaccines targeted at adults, such as the COVID-vaccines), we decided to keep the general a-priori focus on childhood vaccination – however, we of course see the timeliness of and need for qualitative data on COVID-19 vaccination, so we are explicitly probing for themes related to this in the ongoing data collection, and have inserted text in the section on “Data Collection in times of COVID” to clarify this:

	“Additionally, in light of the timely relevance of vaccination research in the context of the ongoing pandemic, we will use these debriefings to discuss emerging topics and potential probing approaches with regards to adult and COVID-19 vaccination and continuously refine data collection tools accordingly.”
Use of ‘eradication’ in the introduction: neither measles nor polio has been eradicated, although on standard definitions both have been eliminated from particular geographic areas.	We have revisited the original text and made edits to enhance clarity: “This drop in confidence has contributed to stagnation or decreases in immunization rates, which in turn has resulted in outbreaks of previously controlled or domestically eliminated diseases such as measles and polio.”
Reviewer 4: Dr. Kim Lavoie Comments	
This was an original, timely protocol paper describing the methods for developing a human-centered intervention to address vaccine hesitancy in the Philippines. This paper was very well written, with a compelling rationale and novel, appropriate, systematic methodological approach for co-developing a intervention with relevant stakeholders. This study has a number of important strengths, including being exceptionally timely (in the context of a worldwide pandemic that has brought vaccine hesitancy to the forefront of public health concerns). This study was exceptionally well reasoned: communicable diseases have been shown to disproportionately affect LMIC’s and vaccine hesitancy has been an important problem in the Philippines in particular. Public trust has been eroded, resulting in outbreaks polio and measles.	We thank the reviewer for pointing out the relevance and timeliness of our work in understanding this complex vaccine hesitancy problem. We appreciate that the reviewer views our methodology as rationale, appropriate and systematic.
The SALUBONG intervention design is rooted in evidence-based behavioural theories (Theory of Planned Behaviour and the Social-Ecological Model) and builds upon long-held local values and customs focused on acceptance, compassion and understanding. The intervention is community-based and being co-developed in collaboration with all key stakeholders (families, healthcare providers, policymakers), which should increase acceptability, uptake and impact. Objectives are clearly defined: they wish to	We thank the reviewer for highlighting the importance of the use of evidence-based theories in understanding locally held values.

develop and pilot a health promotion intervention to address vaccine hesitancy in the Philippines that has the potential to be exported to other settings. Methods are clearly described, systematic, and appropriate. The design includes 5 phases: (1) a preparatory phase that involves conducting interviews with policymakers to understand current challenges; (2) phase 1 (empathize) that involves conducting interviews with hesitant parents of children under 5 (using purposive sampling) and with healthcare workers and community leaders to understand their experiences with childhood vaccination; (3) phase 2 (define and ideate) that involves conducting a records review to identify different categories of participants, followed by focus groups; (4) phase 4 (prototype) which involves developing the first iteration of the intervention; and (5) phase 5 (testing), which involves pilot testing the intervention with a larger sample.	
Methods described at each phase were well defined and appropriate. The authors will use established qualitative methods and analytic strategies, and intervention development stages are in line with established behavioural intervention development models (e.g., MRC, ORBIT). The study populations were well reasoned and carefully selected, and recruitment appears feasible (even in the context of COVID-19). Both qualitative and quantitative analysis methods were well described and appropriate. Sample size estimates were provided and were well justified. Methods are in place to ensure confidentiality and data quality. The integrated knowledge translation approach that involves all relevant stakeholders is evident throughout the project, and should increase acceptance, uptake and efficacy. This study protocol serves as a model for other public health scientists working to change behaviour on a population scale.	We are grateful for the thoughtful observations that pertain to our efforts in presenting our methods in succinct and interesting manner. We value the reviewer's remark that our protocol can be a model for other scientists in understanding and changing behaviors across health discipline.
Some abbreviations are not known and were difficult to follow. Suggest using less abbreviations for less common terms (IDI's, etc).	We acknowledge that there are many acronyms throughout the manuscript – we cut acronyms that were used scarcely throughout the document, and

	we retained those which, in our view, improve clarity and heighten ease of reading.
The authors will contact Department of Health officials to plan aspects of data collection – it is not known if they have already secured their collaboration. This would be important to confirm.	The institute leading the work in the field, the Research Institute for Tropical Medicine, is the research arm of the Philippines Department of Health (DOH) which has facilitated collaboration. Additionally, as the country’s health care system has undergone devolution and decentralization (i.e. creation of provincial, city and regional offices), we have sought formal endorsement from the DOH central office, which has been granted. We clarified this in the text: “All required formal endorsements (i.e., memorandums of agreement) will have been granted prior to data collection.”
Methods are in place to carefully select qualified interviewers who will undergo training in the protocol. However, methods for assessing the success of training and interview fidelity (once the study is running) to ensure that interviewers are respecting interview protocols. This should be added.	Interview fidelity is incorporated into systematic debriefings. We have clarified this in the following text: “The weekly debriefings will also allow us to continuously assess, discuss, and refine study tools, data collection procedures, and emerging issues in data collection as a means to ensure fidelity to the tenets of high-quality interviewing.”
The authors are encouraged to check the Equator Network page to verify if this study needs to adhere to any reporting protocols: https://www.equator-network.org/	We appreciate the feedback; hence, we highlight this suggestion in the new sub-section ‘Ethics and Dissemination’. See edits throughout this section, especially the sentence: “We will adhere to specific reporting guidelines for all publications as applicable, such as the Standards for Reporting Qualitative Research (SRQR) or the Consolidated Criteria for Reporting Qualitative Studies (COREQ).”

VERSION 2 – REVIEW

REVIEWER	Gowda, Charitha Nationwide Children's Hospital, Infectious Diseases
REVIEW RETURNED	17-May-2021
GENERAL COMMENTS	The authors have adequately addressed all of the reviewers' comments.
REVIEWER	Jamrozik, Euzebiusz University of Oxford, Ethox and the Wellcome Centre for Ethics and Humanities
REVIEW RETURNED	12-May-2021

GENERAL COMMENTS

This is a timely project on a crucial topic, and I wish the authors the best of luck with the challenging context in which it will be conducted.

The minor issues raised in response to the first version have been addressed, and the manuscript is worthy of full consideration for publication.